# Alteration of Metabolic Conditions Impacts the Regulation of IGF-II/H19 Imprinting Status in Prostate Cancer

**DOI:** 10.3390/cancers13040825

**Published:** 2021-02-16

**Authors:** Georgina Kingshott, Kalina Biernacka, Alex Sewell, Paida Gwiti, Rachel Barker, Hanna Zielinska, Amanda Gilkes, Kathryn McCarthy, Richard M. Martin, J. Athene Lane, Lucy McGeagh, Anthony Koupparis, Edward Rowe, Jon Oxley, Jeff M. P. Holly, Claire M. Perks

**Affiliations:** 1IGF & Metabolic Endocrinology Group, Translational Health Sciences, Bristol Medical School, Learning & Research Building, Southmead Hospital, Bristol BS10 5NB, UK; mdxkz@bristol.ac.uk (K.B.); mdrmh@bristol.ac.uk (R.B.); hanna.zielinska@hotmail.com (H.Z.); jeff.holly@bristol.ac.uk (J.M.P.H.); claire.m.perks@bristol.ac.uk (C.M.P.); 2Department of Cellular Pathology, North Bristol NHS Trust, Southmead Hospital, Bristol BS10 5NB, UK; alexsewell@doctors.org.uk (A.S.); paida@doctors.org.uk (P.G.); jon.oxley@nbt.nhs.uk (J.O.); 3Department of Pathology, North West Anglia NHS Foundation Trust, Peterborough PE3 9GZ, UK; 4Department of Haematology, Cardiff University, Heath Park, Cardiff CF14 4XN, UK; gilkes@cardiff.ac.uk; 5Department of Surgery, Department of Medicine, Southmead Hospital, Bristol BS10 5NB, UK; Kathryn.mccarthy@nbt.nhs.uk; 6Population Health Sciences, Bristol Medical School, University of Bristol, Canynge Hall, 39 Whatley Road, Bristol BS8 2PS, UK; richard.martin@bristol.ac.uk; 7National Institute for Health Research, Biomedical Research Centre at University Hospitals Bristol and Weston NHS Foundation Trust and the University of Bristol, Biomedical Research Unit Offices, University Hospitals Bristol Education Centre, Dental Hospital, Lower Maudlin Street, Bristol BS1 2LY, UK; 8Bristol Randomised Trials Collaboration, Population Health Sciences, Bristol Medical School, University of Bristol, Canynge Hall, 39 Whatley Road, Bristol BS8 2PS, UK; athene.lane@bristol.ac.uk; 9Supportive Cancer Care Research Group, Faculty of Health and Life Sciences, Oxford Institute of Nursing, Midwifery and Allied Health Research, Oxford Brookes University, Jack Straws Lane, Marston, Oxford OX3 0FL, UK; lmcgeagh@brookes.ac.uk; 10Department of Urology, Bristol Urological Institute, Southmead Hospital, Bristol BS10 5NB, UK; anthony.koupparis@nbt.nhs.uk (A.K.); edward.rowe@nbt.nhs.uk (E.R.)

**Keywords:** IGF2, H19, imprinting, cancer, inflammatory markers

## Abstract

**Simple Summary:**

Insulin-like growth factor II (IGF-II) is a potent growth factor implicated in several cancer types. The IGF-II gene shares its locus with the long non-coding RNA, H19. IGF-II/H19 is an imprinted gene—a phenomenon where only the maternal copy is expressed. The silencing of the paternal copy is lost in many cancer types—including prostate. Our in vitro findings show it is possible to disrupt imprinting through the alteration of metabolic conditions, with changes occurring at the molecular level only. Comparing with prostate tissue samples we additionally found a positive correlation to exist between IGF-II and H19 mRNA expression, which was further confirmed by in silico data from the Cancer Genome Atlas.

**Abstract:**

Prostate cancer is the second major cause of male cancer deaths. Obesity, type 2 diabetes, and cancer risk are linked. Insulin-like growth factor II (IGF-II) is involved in numerous cellular events, including proliferation and survival. The IGF-II gene shares its locus with the lncRNA, H19. IGF-II/H19 was the first gene to be identified as being “imprinted”—where the paternal copy is not transcribed—a silencing phenomenon lost in many cancer types. We disrupted imprinting behaviour in vitro by altering metabolic conditions and quantified it using RFLP, qPCR and pyrosequencing; changes to peptide were measured using RIA. Prostate tissue samples were analysed using ddPCR, pyrosequencing and IHC. We compared with in silico data, provided by TGCA on the cBIO Portal. We observed disruption of imprinting behaviour, in vitro*,* with a significant increase in IGF-II and a reciprocal decrease in H19 mRNA; the increased mRNA was not translated into peptides. In vivo, most specimens retained imprinting status apart from a small subset which showed reduced imprinting. A positive correlation was seen between IGF-II and H19 mRNA expression, which concurred with findings of larger Cancer Genome Atlas (TGCA) cohorts. This positive correlation did not affect IGF-II peptide. Our findings show that type 2 diabetes and/or obesity, can directly affect regulation growth factors involved in carcinogenesis, indirectly suggesting a modification of lifestyle habits may reduce cancer risk.

## 1. Introduction

Prostate cancer is the most frequently diagnosed cancer type and the second major cause of cancer deaths amongst men [1]. Whilst the exact cause of prostate cancer remains unknown, several factors have been found to contribute to disease susceptibility, progression, and prognosis. These include familial history and genetics [2], ethnicity [3], ageing [4] and obesity [5].

In families with a history of prostate cancer, the risk of developing the disease doubles in those with a first-degree relationship [6]. In terms of heritability, it appears that predisposition to the disease is not caused by changes to a single gene; it is due to polygenic mutations [7,8].

Numerous studies have highlighted the disparities in prostate cancer incidence across the globe, focusing on populations in developed and developing countries. The clear link between most is that the incidence of prostate cancer is highest amongst men of Afro Caribbean origin [9,10].

A link exists between obesity and cancer risk [11]. The increased adipose tissue, associated with obesity, produces over 20 hormones and cytokines that can disturb the delicate balance of the cellular environment [12].

In addition to increased cancer risk, obesity also increases susceptibility to type 2 diabetes (T2D). [13]. One key characteristic associated with T2D is hyperglycaemia. As glucose is the main energy source for cells, including cancer cells, an increase in its supply can cause increased proliferative potential [14]. Moreover, a study published in 2019 found a strong link between hyperglycaemia and increased risk of fatal prostate cancer [15].

Insulin-like growth factor II (IGF-II) is a growth factor expressed in high quantities during early embryonic development [16]. Expression continues throughout adulthood, with the liver being the primary site of synthesis [17]. IGF-II plays a critical role in several cellular events, including proliferation and survival [18].

In obese subjects, circulating serum levels of IGF-II are elevated and have shown a positive correlation with increased body mass index (BMI) [19]. Conversely, after weight loss, circulating levels of IGF-II have been shown to drop, regardless of which diet was followed [20].

The IGF-II/H19 gene is located on the short arm of chromosome 11p.15.5 [21]. It is composed of 10 exons and contains 5 promoters. Exons 1–4 and 6 are non-coding. Control of expression takes place from promoters 0 to 4 and is strictly regulated. 128 kb downstream from IGF-II is a long non-coding RNA, H19, which is linked to IGF-II by an imprinting control region (ICR) [22].

IGF-II was the first gene identified to be “imprinted” [23]. This is a heritable epigenetic event whereby, through changes to the DNA structure (such as methylation or histone modification) one parental copy will be silenced or imprinted [24]. The loss of this natural silencing phenomenon (loss of imprinting—LOI) has been identified across a number of cancers, including prostate [25], breast [26], colorectal [27] and lung [28].

Several suggested models for the imprinting mechanism have been proposed, with the most widely accepted being the “enhancer competition model” [29]. This model proposes that the presence of a CCCTC-binding factor (CTCF)—a multi-zinc finger protein (and a transcriptional repressor)—can bind to the unmethylated imprint control region (ICR) embedded within the maternal IGF-II / H19 allele. This impedes IGF-II transcription. Conversely, on the paternal allele, the ICR is hypermethylated. This blocks CTCF from binding, which permits IGF-II to be transcribed. It is thought that loss of imprinting occurs when both parental copies of IGF-II / H19 are hypermethylated at the ICR, resulting in the bi-allelic expression of IGF-II [30].

The function of H19 has been the subject of many conflicting reports. In some cancers, it behaves as an oncogene. For example, in gastric cancer, H19 and an embedded micro RNA (miRNA-675) within its first exon, were found to be increased in tissue and cell lines. This over-expression led to increased cell proliferation and the inhibition of apoptosis [31]. In non-small-cell lung cancer (NSCLC), H19 expression was significantly higher in NSCLC lung tissue compared with normal, being at its highest in those from stage III and IV tumours [32]. In contrast, it has been found to behave as a tumour suppressor in some cancer types, such as colorectal [33] and prostate [34]. The mechanism that dictates H19 behaviour differs between cancer types.

A study conducted in 2011 [35] analysed LOI of IGF-II in human prostates containing cancerous cells. Using qPCR and pyrosequencing, LOI occurred not only in cells adjacent to tumours but also in distant areas within the peripheral region. This suggests a widespread epigenetic field effect in histologically normal tissues.

One study, published in 2012, analysed circulating IGF-II protein levels [36] in patients with a history of PCa. Of 106 (41 patients—radical prostatectomised (RPE)—and 65 controls) IGF-II levels were significantly elevated in the RPE cohort. LOI was also significantly higher in the RPE group (39%) compared to the control (20%). Despite the link between LOI and elevated serum IGF-II in the RPE group, the two were found to be uncoupled. Of the control cohort with LOI (20%), circulating IGF-II levels were found to be very similar to those of the RPE cohort with LOI (39%). Based on these findings, the authors suggested that, under normal conditions, only approximately 35% of the total serum IGF-II is regulated by imprinting [36].

A previous report focused on promoter methylation as a factor contributing to IGF-II transcription [37]. IGF-II mRNA and peptide levels were decreased by 80% of PCa, compared to non-neoplastic adjacent prostate and were independent of LOI status. IGF-II expression in both tumour and adjacent tissue depended on the usage of the IGF-II promoters P3 and P4; decreased IGF-II expression in tumour tissue was strongly related to hypermethylation of these two promoters. The cause of hyper-methylation in this cohort was attributed to cumulative DNA damage, as a result of ageing.

In this study, we examined the in vitro effects of altering metabolic conditions on IGF-II imprinting status, IGF-II / H19 mRNA, and IGF-II peptide levels in the PC3 prostate cancer cell line. In addition, a comparative clinical cohort of prostate cancer tissue was analysed for expression of IGF-II and H19 mRNA using digital droplet polymerase chain reaction (ddPCR) and compared with two larger publicly available patient cohorts from the Cancer Genome Atlas (TGCA). IGF-II imprinting status was also analysed, using pyrosequencing, along with IGF-II localisation and abundance using immunohistochemistry (IHC).

## 2. Results

### 2.1. Identification of Prostate Cell Line Suitability

To enable us to study the effects of metabolic conditions on LOI of IGF-II, we needed to select a suitable cell line; DNA was extracted from 4 immortalised prostate cancer cell lines: PC3, LNCaP, DU145, VcaP and 1 normal prostate cell line: PNT2. These were genotyped using restriction fragment length polymorphism (RFLP) digestion at the APA1 restriction site (rs680); VcaP and PC3 cells were found to be heterozygous, presenting with 2 bands of sizes 292 bp and 229 bp, respectively (Figure 1A). These cell lines were classified as informative and would be suitable for further assessment of imprinting status. Cell lines DU145, PNT2 and LNCaP were found to be homozygous for the SNP at rs680 (i.e., the same snp base at the same location, on both copies of the gene), resulting in a single band pattern and, as such, were deemed non-informative.

### 2.2. Confirmation of IGF-II Imprinting Status in the PC3 and VCaP Cell Lines

The PC3 and VCaP cell lines were then analysed further to establish IGF-II imprinting status; RNA was extracted, converted to cDNA, and genotyped as before using the same primers. The VCaP cell line produced two bands of sizes 292 and 229 bp, confirming bi-allelic expression of IGF-II (IGF-II mRNA transcribed from 2 alleles: maternal and paternal) and, therefore, loss of imprinting (Figure 1B). The PC3 cell line, produced a single band of size 292 bp, indicating mono-allelic expression of IGF-II (IGF-II mRNA transcribed from a single allele) and hence the retention of imprinting status and, therefore, considered to be the most suitable cell line for experiments examining the induced loss of imprinting.

### 2.3. The Impact of Altered Levels of Glucose on IGF-II Imprinting Status

To investigate the effects of altered metabolic conditions upon IGF-II imprinting status, we initially varied glucose levels in the culture media, to mimic conditions of a hyperglycaemic environment. PC3 cells were cultured in 5, 9 and 25 mM glucose media for 6, 24 and 48 h. RNA was extracted, converted to cDNA and genotyped as before. After 48 h, cells exposed to 9 and 25 mM glucose-containing media, showed two bands of 229 and 292 bp. This indicated a change from mono- to biallelic IGF-II expression and, therefore, loss of imprinting (Figure 1C).

### 2.4. The Impact of Altered Levels of Glucose on IGF-II Imprinting Percentage

To further clarify the change in imprinting status, pyrosequencing was used to quantify the loss of imprinting as a percentage. Using the SNP assay facility, within the Qiagen Pyromark Q96 software, it was possible to genotype and calculate imprinting loss using the A/G SNP at rs680. We applied the formula: 2 (A-50), where A > 50, or 2 (50–A), where A < 50. After 6 h, cells cultured in 25 mM glucose showed a significant (*p* < 0.001) decrease in imprinting percentage, compared to those cultured in 5 mM glucose. After 24 h, cells cultured in both 9 and 25 mM glucose showed significant decreases in imprinting percentage (*p* = 0.01 and *p* < 0.001, respectively), compared to those cultured in 5 mM glucose; the effects of 9and 25 mM glucose were enhanced further after 48 h (*p* < 0.001 and 0.001, respectively) when compared to those cultured in 5 mM glucose (Figure 1D).

### 2.5. The Impact of Altered Levels of Glucose on IGF-II mRNA Expression

To ascertain whether changes in imprinting status and percentage, induced by increasing glucose levels, were accompanied by changes in IGF-II (Figure 1E*i*) and H19 (Figure 1E*ii*) mRNA expression, qPCR was used to quantify relative expression of both using GAPDH as the internal control/house-keeping gene. After 6 h, there was a significant decrease in IGF-II mRNA in cells cultured in 9 mM and 25 mM glucose media (*p* = 0.01 and *p* < 0.001 respectively), when compared with the control (5 mM glucose). There were no significant changes to H19 mRNA expression. The same pattern of IGF-II expression was observed after 24 h (*p* < 0.001and 0.001, respectively) when compared with the control and similarly there were no significant changes in H19 mRNA expression. After 48 h, however, there was a significant increase in IGF-II mRNA in cells cultured 25 mM glucose media (*p* = 0.01), when compared with the control. Conversely, a significant decrease in H19 mRNA expression was observed, in cells cultured in 9 mM and 25 mM glucose media (*p* < 0.001 and 0.001, respectively).

### 2.6. Effects of TNFα on IGF-II Imprinting Status

Having established the effects of glucose on IGF-II imprinting, we next investigated the effects of an inflammatory cytokine. PC3 cells were cultured in 5- and 25 mM glucose media and exposed to increasing doses (0 to 10 ng/mL) of tumour necrosis factor-alpha (TNFα) for 24 h. RNA was extracted and genotyped as before. Cells cultured in 5 mM glucose and dosed with 10 ng TNFα showed two bands sized 229 and 292 bp, which indicated a change from mono- to biallelic IGF-II expression, suggesting loss of imprinting (Figure 2A*i*). Cells cultured in 25 mM glucose-containing media and dosed with 1, 5, and 10 ng/mL ng TNFα showed two bands sized 229 and 292 bp (Appendix A). This indicated a change from mono- to biallelic IGF-II expression, suggesting loss of imprinting. The untreated cells, cultured in 25 mM glucose also showed two bands of 229 and 292 bp, confirming the loss of imprinting with raised glucose (Figure 2A*ii*).

### 2.7. Effects of High Dose TNFα on the Degree of IGF-II Imprinting in Cells Cultured in Normal (5mM) Glucose Media

To assess the effects of TNFα upon imprinting percentage, pyrosequencing was used. Cells cultured in 5 mM glucose (Figure 2B*i*) and dosed with 1, 5 and 10 ng/mL all showed significant (*p* < 0.001) decreases in imprinting percentage, compared to controls. Cells cultured in 25 mM glucose (Figure 2B*ii*) and dosed with 5 and 10 ng/mL TNFα both showed highly significant decreases in imprinting percentage (*p* < 0.001 and 0.001, respectively) when compared to controls.

### 2.8. Effects of TNFα on IGF-II mRNA Expression in Cells Cultured in Normal (5 mM) Glucose Media

To ascertain whether changes in imprinting status and percentage, induced by TNFα, were accompanied by changes in IGF-II and H19 mRNA expression, qPCR was conducted. Cells cultured in 5 mM glucose media showed a significant decrease in IGF-II mRNA expression when dosed with 1 and 5 ng/mL TNFα (*p* = 0.01 and 0.05, respectively), compared with the untreated control. In contrast, cells treated with 10 ng/mL TNFα showed a significant (*p* = 0.01) increase in IGF-II mRNA expression, when compared with the control. No significant fold changes in H19 mRNA expression were recorded (Figure 2C*i*).

Cells cultured in 25mM glucose showed no significant changes in IGF-II mRNA when compared to the untreated control. Conversely, a significant decrease (*p* = 0.01) in H19 mRNA was observed in cells treated with 1 ng/mL TNFα, compared to the un-dosed control. A significant increase (*p* < 0.001) in H19 mRNA was observed in cells treated with 10 ng/mL of TNFα when compared with the un-dosed control (Figure 2C*ii*).

### 2.9. Effects of TNFα on Levels of Secreted IGF-II Peptide

Having observed the influence of TNFα on IGF-II mRNA expression, we next investigated whether this translated into the secreted peptide. Conditioned media from the TNFα dosing experiments was collected and quantified for peptide using radioimmunoassay (RIA). Cells cultured in 5 mM glucose and exposed to increasing doses of TNFα (0, 1, 5 and 10 ng/mL) for 24 and 48 h, showed no significant changes to secreted IGF-II peptide (Figure 2D*i*). This was also true of cells cultured in 25 mM glucose, exposed to identical doses of TNFα after identical time points (Figure 2D*ii*).

### 2.10. IGF-II Imprinting Status Does Not Significantly Vary Between Benign and Malignant Paired Prostate Tissue Samples

After examining imprinting behaviour in vitro, we next compared imprinting status in prostate tissue samples. For this, prostate tissue from a large clinical cohort, previously collected for the PrEvENT study (ISRCTN reference number ISRCTN99048944) [38] was used. Total RNA was extracted from formalin-fixed paraffin-embedded (FFPE) specimens and pyrosequenced to clarify imprinting status (defined by Qiagen Pyromark Q96 software), and imprinting loss as a percentage. 84 patients from the PrEvENT cohort had representative paired samples of benign and malignant tissue. Peak heights at A/G rs680 SNP were converted into percentages. Tissue was typed as eith“r "maintaining imprinting sta”us" (MOI) “r "loss of imprinting sta”us" (LOI). Figure 3A depicts the total cohort with proportional representation of each tissue pairing; out of 84 patients 64 presented with no change of MOI status (in the benign sample compared to its cancer counterpart), 6 presented with a change in status from MOI to LOI, 7 presented with a change in status from LOI to MOI and 7 presented with no change of LOI status. Figure 3B shows the percentage imprinting within each genotype grouping. The group containing the 6 paired samples, with LOI in cancer compared with MOI in the benign tissue, showed a significant decrease (*p* < 0.001) in imprinting percentage. The group containing the 7 paired samples, with MOI in cancer compared with LOI in the benign tissue showed a significant increase (*p* < 0.001) in imprinting percentage.

### 2.11. There Is a Positive Correlation Between IGF-II and H19 mRNA Expression in Benign and Malignant Prostate Tissue

We used ddPCR to quantify relative expression of mRNA for IGF-II and H19 in samples from the PrEvENT cohort, in both benign and malignant tissue, using housekeeping gene G2/M phase-specific E3 ubiquitin-protein ligase (G2E3). There was no difference in absolute expression of IGF-II or H19 between benign and malignant tissue. However, we did observe a positive correlation between IGF-II and H19 expression; with a significant weak positive correlation (Pearson: R = 0.28, *p* = 0.01) in the benign tissue (Figure 4A*i*) and a stronger positive correlation in malignant prostate tissue (R = 0.67, *p* < 0.0001) (Figure 4A*ii*).

After quantifying IGF-II and H19 mRNA expression in the PrEvENT cohort, we utilised the cBioPortal for Cancer Genomics website to assess whether co-expression of IGF-II and H19 mRNA existed, in a larger prostate cancer cohort, as well as another hormone-responsive cancer type: breast.

In the prostate cohort—provided by the Cancer Genome Atlas (TGCA)—a strong positive correlation (Pearson: 0.6) was seen in the co-expression of H19 and IGF-II (*p* < 0.001) with 489 patients (Figure 4B*i*). Similarly, in the TGCA breast cancer cohort (Figure 4B*ii)*, a strong positive correlation (Pearson: 0.64) was also seen in the co-expression of H19 and IGF-II (*p* < 0.001) with 994 patients.

### 2.12. There is a Positive Correlation between IGF-II mRNA and the Degree of Imprinting in Benign and Malignant Prostate Tissue

To identify whether IGF-II mRNA expression correlates with imprinting percentage in the PrEvENT cohort, ddPCR data was plotted against imprinting percentages; a significant (*p* = 0.05) positive correlation (R = 0.328) was found in benign prostate tissue (Figure 4C*i*), which was marginally weaker in malignant tissue (*p* = 0.05 and R = 0.224) (Figure 4C*ii*).

### 2.13. Levels of IGF-II Peptide Do not Differ between Benign and Malignant Prostate Tissue

Having analysed IGF-II mRNA expression in the PrEvENT cohort, we then assessed the abundance of peptides; sections were taken from FFPE tissue specimens and stained for IGF-II peptide, using immunohistochemistry (IHC). The Allred scores of IGF-II peptide abundance in benign and cancerous prostate tissue (*n* = 94) showed no significant differences in stain intensity and proportion (Figure 5A). Micrograph images (Figure 5B) illustrate strong IGF-II peptide staining in cancer (image A) and benign (image D) tissues, and weak staining in cancer (C) and benign (B) tissues.

## 3. Discussion

Loss of imprinting in the growth factor, IGF-II, is a phenomenon common to many different cancer types [39]. The causes and consequences remain unclear. If modifiable behaviours can be shown to influence its occurrence, this could have important implications for understanding the development and progression of cancers.

In this study, we have demonstrated that under either hyperglycaemic or inflammatory metabolic conditions, loss of IGF-II imprinting can be induced in vitro. In high glucose (i.e., above the healthy range of 4 to 5.9 mM/L for an adult according to NICE guidelines (The National Institute for Health and Care Excellence, 2012) or increased inflammatory (TNFα 10 ng/mL) conditions, IGF-II loses imprinting with the introduction of gradual expression of the silenced copy. However, the marked increase in mRNA did not translate into an increased expression of IGF-II peptide.

Whilst the exact mechanism of imprinting loss is unclear it may be hypothesised that exposure to elevated glucose conditions and/or inflammatory cytokines may disrupt the methylation patterns in and around the IGF-II/H19 locus. Our in vitro data indicated that LOI was accompanied by an increase in expression of IGF-II and a reciprocal reduction in expression of H19 consistent with the "enhancer competition model", first posited in 1993 by Bartolomie et al. [29]. Further pyrosequencing to analyse methylation patterns may provide further clarification. Exposure of bovine kidney epithelial cells to butyrate also resulted in increased expression of IGF-II and reduced expression of H19, although the mechanism, in that case, may be more straightforward, as butyrate is a known epigenetic regulator due to its activity as an inhibitor of histone deacetylases, increasing global acetylation [40]. Oxidative stress can also induce loss of imprinting, in both benign and malignant prostate tissue, by disrupting the expression of CTCF and its binding to H19 and the ICR [41].

In the PrEvENT cohort, most matched benign and malignant specimens showed no change in imprinting status, with MOI being most prevalent in both tissue types. This disagrees with findings that have shown LOI to be more common in prostates associated with cancer [42], however, our results were limited by the relatively small cohort size.

In contrast to our in vitro findings, we demonstrated a positive correlation between IGF-II and H19 mRNA expression in the prostate tissue; whilst this was weaker in benign tissue, the stronger positive correlation—seen in malignant tissue—concurred with that of the larger TGCA prostate and breast cancer cohorts.

Our in vitro findings may be explained by the use of the PC3 cell line—a cultured monolayer of cells, of a single cell type—compared to the use of clinical tissue specimens, which conversely, are greatly heterogeneic in nature, as shown by Küffer et al. in their work [37].

Interestingly, IGF-II / H19 LOI is also found in normal tissue and not just cancer, where frequently it is also not related to expression levels. For example, 22% of normal infants have IGF-II LOI with no changes in expression levels in the majority [43]. We suggest that imprinting is only one factor in regulating expression and may not be that important in human tissues; IGF-II and H19 share a common enhancer 3′-downstream from H19 [44] that could be a more important determinant of expression of both, rather than imprinting controls.

The key finding that bridges our in vitro with in vivo findings is that there was no measurable change in IGF-II peptide. The lack of correlative findings may be explained by a deficiency of RNA binding proteins. One such group of binding proteins are the insulin-like growth factor 2 messenger-RNA binding proteins (IMPs). These affect the processing of IGF-II mRNA at multiple levels including translation. Of the three IMPs (IMP1, IMP2 and IMP3) identified [45], IMP3 has been shown to activate translation of mRNA by binding to the coding regions of IGF-II [46]. Under our simulated in vitro conditions, inhibition or insufficient synthesis of IMP3 may have occurred and further quantification of IMP3 would address this issue.

In vivo, a deficiency of RNA binding proteins may be also contributory, but an additional factor may be that circulating IGF-II peptide is unstable and may be susceptible to degradation [47]. To combat this, it binds with specific binding proteins—insulin-like growth factor binding proteins (IGFBPs). There are six IGFBPs: IGFBP1-6 and they bind to IGF-I and -II with high affinity [48]. More specifically, IGFBP2 binds to IGF-II with very high affinity [49]. Therefore, the inconsistency of IGF-II staining is likely due to the presence of peptide bound to IGFBP-2.

Different associations have been reported between IGF-II LOI, mRNA and peptide levels and this reflects the limited number of studies and the use of diverse methodologies and assays. For example, one report in men with prostate cancer found no relationship between IGF-II LOI status in circulating blood and serum IGF-II levels, whereas a relationship between the two existed in men without prostate cancer [36]. In contrast, IGF-II peptide was increased in triple-negative breast tumours with LOI [50] and a relationship between IGF-II levels in blood was found with LOI in gastric cancers [51].

## 4. Methods

### 4.1. Prostate Cell Lines

Immortalised prostate cancer cell lines PC3, LNCaP, DU145 and VCaP, and a normal prostate epithelial cell line—PNT2, were purchased from the American Type Culture Collection (ATCC, Manassas, VA, USA). PC3, VCaP and DU145 were grown and maintained in Dulbecco’s modified Eagles Medium (DMEM, BioWittaker, Verviers, Belgium) supplemented with 10% foetal bovine serum (FBS, Gibco, Paisley, UK) and 1% L-glutamine solution (2 mM; Sigma, Dorset, UK). LNCaP and PNT2 cell lines were grown and maintained in Roswell Park Memorial Institute (RPMI-1640, BioWittaker, Verviers, Belgium) growth medium supplemented with 10% FBS and 1% L-glutamine solution, as before. Cells were incubated at 37 °C, in a humidified 5% carbon dioxide environment.

### 4.2. Isolation of Nucleic Acids

DNA and total RNA were isolated using DNAzol^®^ and RNAzol^®^ (Invitrogen, Thermo Fisher Scientific, Paisley, UK) reagents respectively, according to the manufacturer’s instructions.

DNA was resuspended in 40mM NaOH (Merck Life Science, Dorset, UK) and RNA was resuspended in RNase-free water; both were quantified using a NanoPhotometer^TM^ (Implen, München, Germany). Extracts were stored at –20 °C and –80 °C, respectively.

### 4.3. Preparation of cDNA

RNA was treated with DNase I (Sigma-Aldrich, Gillingham, UK) prior to cDNA conversion, using the High-Capacity RNA-to-cDNA Kit^®^ (Applied Biosystems, Foster City, CA, USA). cDNA was made from 2 µg RNA, as per kit protocol. The resulting product was stored at –20 °C.

### 4.4. Genotyping of Cell Lines, Using Restriction Fragment Length Polymorphism (RFLP) Analysis

Cell DNA and RNA (cDNA) was genotyped by exploiting the presence of a single nucleotide polymorphism (SNP) in exon 7 of IGF-II, at restriction site 680 (APA1). Cell lines that were heterozygous for this SNP at the DNA level were then used to identify whether IGFII mRNA (cDNA) was transcribed from one or two alleles. The RFLP process was conducted in two stages: The first amplified IGF-II product via PCR, using the following primer sequences: F cttggactttgagtcaaattgg and R ggtcgtgccaattacatttca (Sigma-Aldrich). The second stage used an APA1 digestion enzyme (catalogue no. FD1414, Thermo Fisher Scientific), as per kit protocol, to digest product from stage 1 and “cut” at rs680. The resulting products were run on a 2% agarose (Bio-Rad, Hercules, CA, USA) gel and visualised using a trans-illuminator (Bio-Rad).

### 4.5. Dosing Experiments

For all experiments, cells were seeded at a density of 1 × 10^6^ million cells per T25 flask (Greiner Bio-one, Gloucestershire, UK) in normal glucose-containing medium (1000 mg/L–5 mM). After 24 h, media was exchanged for serum-free media (SFM) supplemented with sodium bicarbonate (1 mg/mL) (Sigma-Aldrich), bovine serum albumin (0.2 mg/mL, Sigma-Aldrich) and apo-transferrin (0.01 mg/mL) (Sigma-Aldrich), with normal glucose (5 mM), moderate glucose (9 mM) and high glucose SFM (25 mM).

At 6, 24 and 48 h intervals, a flask from each different glucose concentration was removed and processed for RNA extraction. At the end of each time point, there were 3 flasks containing 5 mM, 9 mM and 25 mM SFM.

For TNFα dosing, cells were seeded in 5 mM glucose-containing DMEM. After 24 h, media was replaced with 5 mM and 25 mM glucose SFM. 24 h later, cells were dosed with or without TNFα at 1, 5 and 10 ng/mL, respectively. Cells were processed for RNA and DNA extraction.

### 4.6. Quantitative PCR (qPCR)

qPCR was performed using SYBR Green (Applied Biosystems) on a StepOne Real-Time PCR machine (Thermo Fisher Scientific). GAPDH was used as an internal control. Primer sequences were as follows: GAPDH *F*-CATCTTCTTTTGCGTCGCCA and *R*-TTAAAAGCAGCCCTGGTGACC, IGF-II *F*- GAGCTCGAGGCGTTCAGG and *R*-GTCTTGGGTGGGTAGAGCAATC and H19 *F*-CGGAACATTGGACAGAAG and *R*-GGCGAGGCAGAATATAAC (Sigma-Aldrich). Quantitation of product was conducted using the Quantitation Comparative CT function within StepOne software (ThermoFisher Scientific). Relative expression was calculated using the Pfaffl method [52].

### 4.7. Pyrosequencing

Preparation of cDNA for pyrosequencing was conducted as previously described [53], with the nested PCR step scaled up four-fold to yield 40 µl product as per assay requirements defined by the Qiagen Pyromark Q96 protocol. Two sets of primers were used, with the following sequences: Set 1) F- ATCGTTGAGGAGTGCTGTTTCC & R- GAGCCAGTCTGGGTTGTTGC; Set 2/Nested) F- AGTCCCTGAACCAGCAAAGAG & R-biotin-TCGGATGGCCAGTTTACC; the pyrosequencing primer sequence was as follows: AGCAAAGAGAAAAGAAGG (Eurofins). The PCR product was prepared for pyrosequencing using Qiagen PyroMark Gold Q96 reagents and buffers (Qiagen, Manchester, UK) and sequenced on a PyroMark Q96 machine (Qiagen) by kind permission of Cardiff University Hospital, Cardiff, Wales – UK and North Bristol NHS Trust, Department of Pathology, Southmead Hospital, Bristol, UK. Peak heights and genotypes were quantified by the Qiagen Pyromark software SNP application; peak heights at the rs680 A/G SNP were converted to imprinting percentages using the following formulae: Where A > 50%: 2 (A-50). Where A < 50%: 2 (50-A).

### 4.8. Radioimmunoassay (RIA)

The concentration of IGF-II peptide was quantified in-house, using radioimmunoassay (RIA) as previously described [54]. This method quantifies all forms of IGF-II peptide, including those that have undergone fragmentation.

### 4.9. Immunohistochemistry (IHC)

Using a microtome (Leica), 8 µm thick slices were taken from formalin-fixed paraffin-embedded (FFPE) prostate tissue blocks and mounted on Tomo microscope slides (Matsunami, Bellingham, WA, USA). Slides were then stained for IGF-II peptide using an IGF-II antibody (AbCam, Cambridge, UK) at a dilution of 1:600, on a Ventana BenchMark ULTRA™ machine (Roche, Oro Valley, AZ, USA). Slides were scored by two pathologists using a modified version of the Allred system, combining the proportion of tissue stained (a scale of 1–5) with the staining intensity (a scale of 1–3) to give a score out of 8 [55]. Currently, there is no standardized method of scoring prostate tissue.

### 4.10. Digital Droplet PCR (ddPCR)

Primers and Taqman probes were purchased from Bio-Rad and Thermo Fisher Scientific; to optimize the detection of total IGF-II and H19 transcript expression with ddPCR, we initially performed an annealing temperature gradient test for our assay using the prostate cancer cell line PNT2 total RNA. At 55 °C, annealing temperature for ddPCR reactions total IGF-II and H19 amplicon-containing droplets showed the best separation between positive and negative clusters. DDPCR samples for total IGF-II and H19 were set up with 16ng cDNA samples from individual patients; a stock mix was prepared which contained 10 µL ddPCR Supermix for Probes, no dUTP (Bio-Rad), 500 nM of each forward and reverse primer (FP and RP) and 250 nM probe (FAM and HEX). Each reaction volume totalled 20 µL. Droplets were generated with 70 µL oil using a QX200 droplet generator (Bio-Rad). Amplification was performed at 95 °C, 10 min; followed by 40 cycles of 94 °C, 30 s and 55 °C, 1 min using a C1000 Touch thermocycler (Bio-Rad). After amplification, the droplets were read on a QX200 droplet reader (Bio-Rad) and analysed with QuantaSoft software V1.7.4 (Bio-Rad), which converted droplets to copies/ng RNA based on the RNA concentration and the total volume added to the reaction dependent on Poisson distribution. For normalized expression, we used the ratio of IGF-II/H19 concentration to an internal/housekeeping control G2E3 concentration. Raw fluorescence amplitude data were extracted from the droplet reader. We excluded/repeat samples which had too many positive or negative droplets or less than 10,000 droplets.

### 4.11. Use of cBioPortal for Cancer Genomics to Examine co-expression of IHG-II & H19 mRNA

One representative Cancer Genome Atlas cohort was selected from the list of breast and prostate cancer studies. Each cohort was assessed for the availability of IGF-II and H19 mRNA co-expression data, which had been gathered using the Illumina HiSeq sequencing system.

### 4.12. Statistical Analysis

Analysis of variance (ANOVA) was used for in vitro data, where comparative control experiments were conducted; error bars show the standard error of the mean. A two-tailed *t*-test was used for in vivo data, to compare the differences between two groups. For correlation analyses, the Pearson correlation coefficient was calculated, with results being expressed as an R-value. P values are denoted as follows: * *p* = 0.05, ** *p* = 0.01 and *** *p* < 0.001.

## 5. Conclusions

Our study illustrated that an artificially simulated inflammatory environment does induce loss of imprinting in IGF-II, causing upregulation of IGF-II and downregulation of H19 mRNA. The potential cause may be due to disruption of methylation patterns throughout both copies of the gene, inducing partial expression of the usually silenced paternal copy. The increase in IGF-II mRNA did not equate to elevated protein expression which may be due to impaired RNA-binding protein synthesis. Matched benign and malignant specimens from the PrEvENT cohort primarily retained imprinting status, although a small subset with LOI did exhibit reduced imprinting. A positive correlation was seen between IGF-II and H19 mRNA expression, which concurred with findings of larger TGCA cohorts and other hormone-responsive cancer types (breast). This positive correlation did not affect IGF-II peptide. Our in vitro findings are of utmost relevance to indicate that type 2 diabetes and/or obesity directly affect the regulation of growth factors involved in carcinogenesis. Furthermore, our data suggest that LOI does not necessarily coincide with the development of cancer or translate into more IGF-II peptide, but its actual consequences may only be fully understood when the complex interactions between all the products of the gene locus, including long non-coding RNA, microRNA and antisense transcripts, are fully characterised.

## Figures and Tables

**Figure 1 cancers-13-00825-f001:**
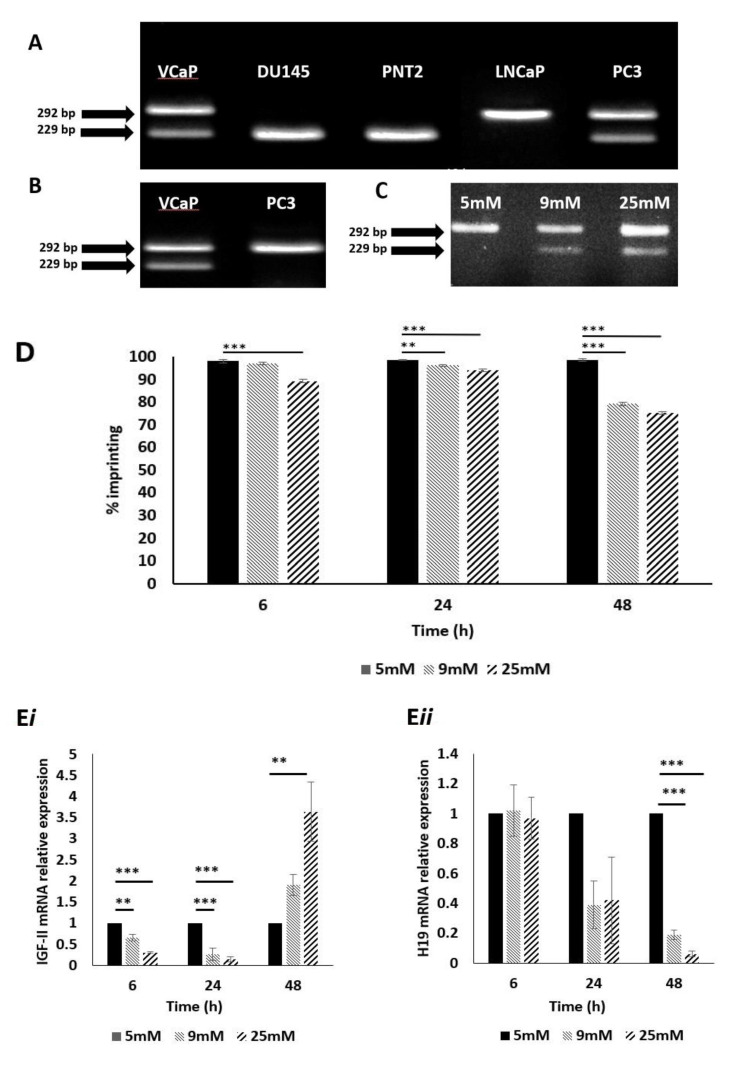
Genotyping, IGF-II allelic expression and imprinting behaviour after treatment with glucose, in prostate cancer cell lines using APA1 RFLP, PIE and qPCR analyses. (**A**) Gel image depicting genotype results of cell lines VcaP, DU145, PNT2, LNCaP and PC3. VcaP and PC3 showed two bands, sized 292- and 229 bp. This meant that these cells were heterozygous for the SNP at rs680 and, therefore, informative. (*n* = 3). (**B**) Gel image depicting IGF-II allelic expression. The VcaP cell line showed two bands, sized 292 and 229 bp, signifying bi-allelic expression and, therefore, loss of imprinting of IGF-II. The PC3 cell line showed a single band sized 292 bp, indicating mono-allelic expression of IGF-II and, therefore, a retention of imprinting status. (*n* = 3). (**C**) Gel image depicting the effects of varied glucose concentration on IGF-II allelic expression in the PC3 cell line after 48 h. Cells cultured in 9- and 25 mM glucose showed two bands sized 229 and 292 bp, indicating a change from mono- to biallelic IGF-II expression and, therefore, loss of imprinting. (*n* = 3). (**D**) Imprinting percentage change of PC3 cells after exposure to glucose for 6, 24 and 48 h. After 6 h, cells cultured in 25 mM glucose showed a highly significant (*p* < 0.001) decrease in imprinting percentage compared to those cultured in 5mM glucose. After 24 h, cells cultured in 9 and 25 mM glucose showed significant decreases in imprinting percentage (*p* = 0.01 and *p* < 0.001, respectively), when compared to those cultured in 5 mM glucose. After 48 h, cells cultured in 9 and 25 mM glucose showed highly significant decreases in imprinting percentage (*p* < 0.001 and <0.001 respectively), when compared to those cultured in 5 mM glucose. (*n* = 3). I IGF-II and H19 mRNA relative expression in the PC3 cell line, after exposure to increased glucose levels; a significant increase in IGF-II mRNA expression (*i*) was observed after 6, 24 and 48 h in 9 and 25 mM glucose media. A significant decrease in H19 mRNA expression (*ii*) was observed after 48h in 9- and 25 mM glucose media. ** *p* = 0.01 and *** *p* < 0.001)

**Figure 2 cancers-13-00825-f002:**
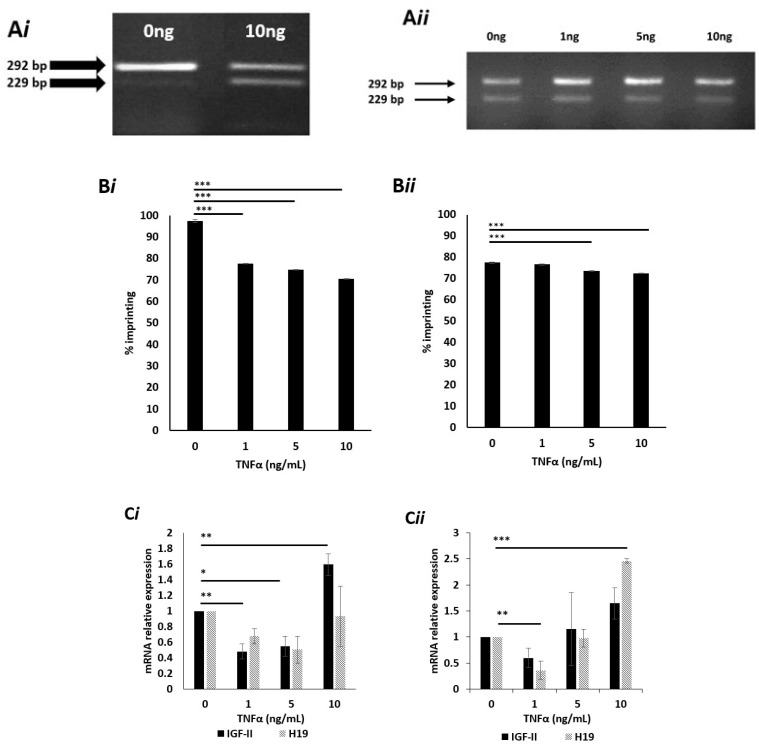
IGF-II allelic expression, imprinting behaviour and peptide expression after treatment with TNFα, in prostate cancer cell lines using APA1 RFLP, PIE, qPCR and RIA analyses. (**A**) Gel image depicting the effects of TNFα upon IGF-II allelic expression in the PC3 cell line, after 24 h exposure. Cells cultured in 5mM (*i*) glucose and dosed with 10 ng/mL TNFα showed two bands, sized 229 and 292 bp, indicating a change from mono- to biallelic IGF-II expression and, therefore, loss of imprinting. Cells cultured in 25 mM (*ii*) glucose and dosed with 0, 1, 5 and 10 ng/mL TNFα showed two bands sized 229 and 292 bp, indicating a change from mono- to biallelic IGF-II expression and, therefore, loss of imprinting. (*n* = 3). (**B**) Imprinting percentage change of PC3 cells after exposure to TNFα for 24 h in 5 mM (*i)* and 25 mM (*ii*) glucose. Cells cultured in 5 mM glucose media (*i*) and dosed with 1, 5 and 10 ng/mL TNFα all showed highly significant (*p* < 0.001) decreases in imprinting percentage; those cultured in 25 mM glucose media (*ii*) and dosed with 5 and 10 ng/mL TNFα, also showed highly significant (*p* < 0.001) decreases in imprinting percentage when compared with the control. (*n* = 3). (**C**) IGF-II and H19 mRNA relative expression in the PC3 cell line, after treatment with TNFα in 5 mM (*i*) and 25 mM (*ii*) glucose for 24h, using qPCR; (*i*) a significant increase in IGF-II was observed in cells dosed with 1, 5 and 10 ng/mL TNFα. No significant fold changes in H19 mRNA were observed (*n* = 3). (*ii*) No significant fold changes in IGF-II mRNA were observed. A significant decrease in H19 was observed in cells dosed with 1ng/mL TNFα, whilst a significant increase was observed in those dosed with 10ng/mL TNFα. (**D**) The effects of glucose and TNFα upon IGF-II peptide after 24 and 48 h exposure to 5 mM (*i*) and 25 mM glucose (*ii*) combined with increasing doses of TNFα, quantified using RIA. There were no significant changes in the quantities of IGF-II peptide. after 24 and 48 h exposure to 5 (*i*) or 25 mM (*ii*) glucose, when combined with increasing doses of TNFα. (*n* = 3). (* *p* = 0.05, ** *p* = 0.01 and *** *p* < 0.001)

**Figure 3 cancers-13-00825-f003:**
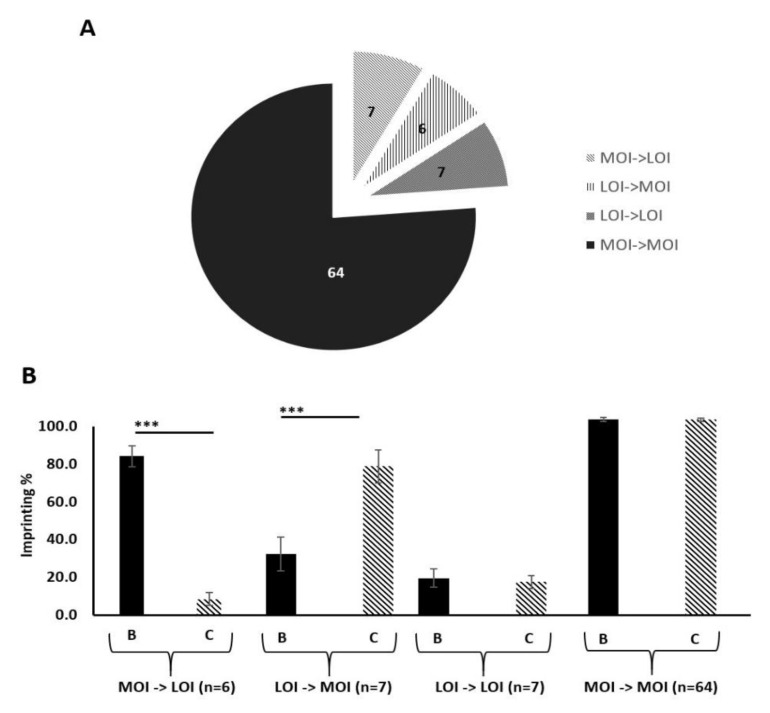
Imprinting status of paired benign and malignant prostate tissue samples from the PrEvENT cohort. (**A**) Cohort divided by imprinting status change, moving from benign to corresponding malignant counterpart. Of the total cohort (84 tissue pairs), 64 showed no change of maintenance of imprinting (MOI) from benign to malignant; 7 showed no change of loss of imprinting (LOI), from benign to malignant; 6 showed a change from LOI to MOI and 7 showed a change from MOI to LOI. (**B**) Imprinting status sub-divisions of PrEvENT cohort, with changes in imprinting percentage. A significant decrease (*p* < 0.001) in imprinting percentage was seen in samples where imprinting status changed from MOI to LOI (*n* = 7); conversely, a significant increase (*p* < 0.001) in imprinting percentage was seen in samples where imprinting status changed from LOI to MOI (*n* = 6). (B = benign tissue, C = cancer tissue) (*** *p* < 0.001)

**Figure 4 cancers-13-00825-f004:**
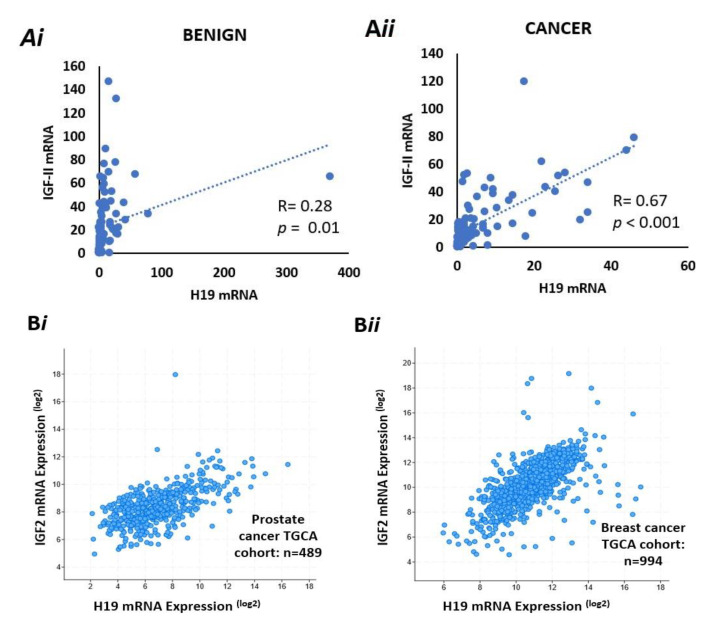
Co-expression of IGF-II & H19 mRNA and IGF-II & % imprinting in benign and prostate cancer tissue sections. (**A**) Co-expression of H19 & IGF-II mRNA in benign (*i)* and cancer (*ii*) derived from FFPE prostate tissue from the PrEvENT cohort. Pearson’s correlation coefficient analysis found a significant (*p* = 0.01) weak positive (R = 0.28) correlation between H19 and IGF-II mRNA expression in benign prostate tissue; a significant (*p* < 0.001) stronger (R = 0.67) positive correlation was found in cancer tissue (*n* = 95). (**B)** Metabric data from The Cancer Genome Atlas (TGCA) showed strong positive correlations between co-expression of H19 & IGF-II mRNA in (*i*) prostate (R = 0.6, *p* < 0.001) (*i*) and (*ii*) breast (R = 0.64, *p* < 0.001 cancer. (**C**) Imprinting percentage of IGF-II gene plotted against IGF-II mRNA in (*i*) benign and (*ii*) cancer prostate tissue. A significant positive correlation was seen between IGF-II imprinting percentage and mRNA expression in both tissue types. (*n* = 84).

**Figure 5 cancers-13-00825-f005:**
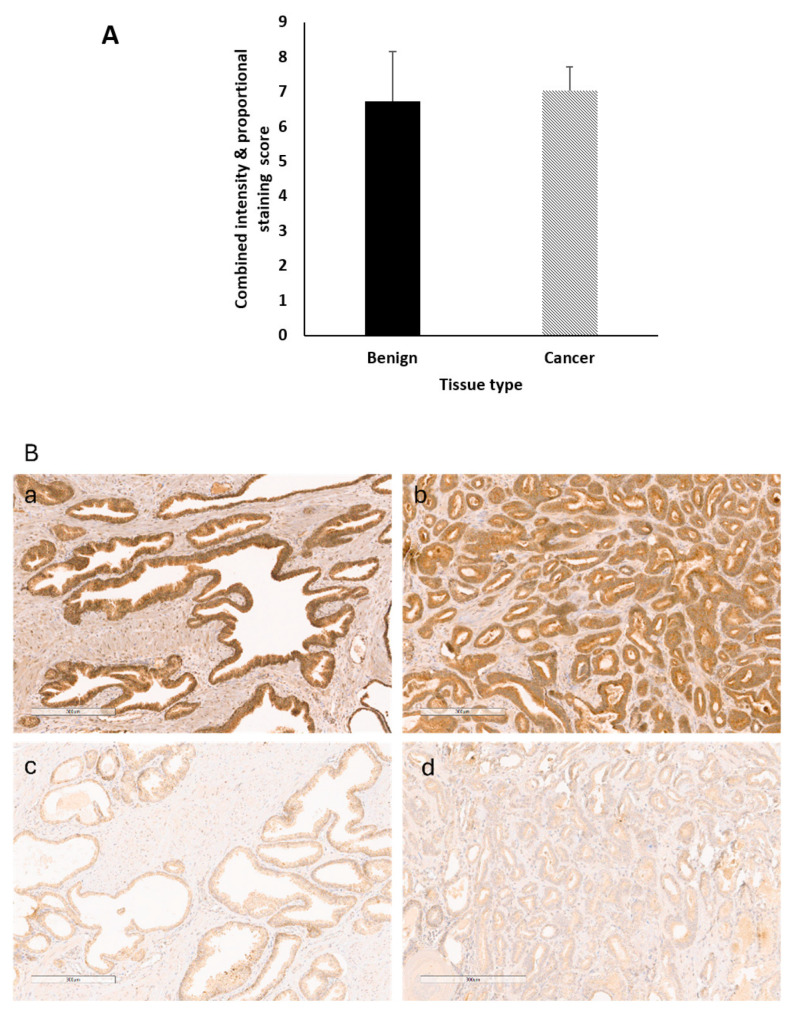
IHC staining of IGF-II peptide in benign and malignant prostate tumour tissue. (**A**) IHC staining of IGF-II peptide in FFPE prostate tumour tissue using the Ventana Benchmark Ultra machine. There were no was no significant differences in staining scores between malignant and benign. (**B**) Immunohistochemical analysis of IGF-II in two individuals, one with strong IGF-II staining in both benign (**a**) and cancer (**b**) tissue and one with weak IGF-II staining in both benign (**c**) and cancer (**d**) tissue. Scale bar = 300 um.

## Data Availability

The data used throughout the study are available on request from the corresponding author.

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
