# Peer review of "Alteration of Metabolic Conditions Impacts the Regulation of IGF-II/H19 Imprinting Status in Prostate Cancer"

_cancers, 2021, doi:10.3390/cancers13040825_

Round 1

Reviewer 1 Report

The manuscript is original in its content and faces an  interresting topics; in my opinion the reference list should be improve with more recent papers.

Furthermore, the introduction should be more detailed on prostate cancer and its mechanisms, adding some schemes if necessary.

There is a typo error, line 80 a comma instead of a dot.

Author Response

Reviewer 1 –

  • “The manuscript is original in its content and faces interesting topics; in my opinion the reference list should be improved with more recent papers.”

As requested, we have replaced several references with more recent citations, as indicated below:

Original reference: 4 (line 59)

Bell, K.J., et al., Prevalence of incidental prostate cancer: A systematic review of autopsy studies. Int J Cancer, 2015. 137(7): p. 1749-57.

New reference: 4 (line 63)

Rawla, P., Epidemiology of Prostate Cancer. World J Oncol, 2019. 10(2): p. 63-89.

Original reference: 5 (line 60)

Jones, A.L. and F. Chinegwundoh, Update on prostate cancer in black men within the UK. Ecancermedicalscience, 2014. 8: p. 455. (line 60)

New reference: 3 (line 63)

Taitt, H.E., Global Trends and Prostate Cancer: A Review of Incidence, Detection, and Mortality as Influenced by Race, Ethnicity, and Geographic Location. Am J Mens Health, 2018. 12(6): p. 1807-1823

Original reference: 6 (line 61)

Allott, E.H. and S.D. Hursting, Obesity and cancer: mechanistic insights from transdisciplinary studies. Endocr Relat Cancer, 2015. 22(6): p. R365-86

New reference: 11 (line 73)

Avgerinos, K.I., et al., Obesity and cancer risk: Emerging biological mechanisms and perspectives. Metabolism, 2019. 92: p. 121-135.

Original reference: 7 (line 63)

Booth, A., et al., Adipose tissue, obesity and adipokines: role in cancer promotion. Horm Mol Biol Clin Investig, 2015. 21(1): p. 57-74. (line 63)

New references: 12 & 13 (line 75):

12Lengyel, E., et al., Cancer as a Matter of Fat: The Crosstalk between Adipose Tissue and Tumors. Trends Cancer, 2018. 4(5): p. 374-384.

13Quail, D.F. and A.J. Dannenberg, The obese adipose tissue microenvironment in cancer development and progression. Nat Rev Endocrinol, 2019. 15(3): p. 139-154.

Original reference: 8 (line 65)

Garg, S.K., et al., Diabetes and cancer: two diseases with obesity as a common risk factor. Diabetes Obes Metab, 2014. 16(2): p. 97-110.

New references: 14 & 15 (line 77)

14Srivastava, S.P. and J.E. Goodwin, Cancer Biology and Prevention in Diabetes. Cells, 2020. 9(6).

15Goto, A., et al., Diabetes and cancer risk: A Mendelian randomization study. Int J Cancer, 2020. 146(3): p. 712-719

Original reference: 9 (line 67)

Chang, S.C. and W.V. Yang, Hyperglycemia, tumorigenesis, and chronic inflammation. Crit Rev Oncol Hematol, 2016. 108: p. 146-153.

New reference: 16 (line 79)

Ramteke, P., et al., Hyperglycemia Associated Metabolic and Molecular Alterations in Cancer Risk, Progression, Treatment, and Mortality. Cancers (Basel), 2019. 11(9).

Original reference: 13 (line 72)

Resnicoff, M., et al., The insulin-like growth factor I receptor protects tumor cells from apoptosis in vivo. Cancer Res, 1995. 55(11): p. 2463-9.

New reference: 20 (line 85)

Simpson, A., et al., Insulin-Like Growth Factor (IGF) Pathway Targeting in Cancer: Role of the IGF Axis and Opportunities for Future Combination Studies. Target Oncol, 2017. 12(5): p. 571-597.

Original reference: 28 (line 103)

Yoshimizu, T., et al., The H19 locus acts in vivo as a tumor suppressor. Proc Natl Acad Sci U S A, 2008. 105(34): p. 12417-22.

New reference: 35 (line 116)

Yuan, L., et al., Long noncoding RNA H19 protects H9c2 cells against hypoxiainduced injury by activating the PI3K/AKT and ERK/p38 pathways. Mol Med Rep, 2020. 21(4): p. 1709-1716.

  • “Furthermore, the introduction should be more detailed on prostate cancer and its mechanisms, adding some schemes if necessary”.

Thank you for this comment. I have expanded on the following information (lines 59-60):

‘…These include familial history and genetics [2], obesity [3], aging [4] and ethnicity [5].’

…These include familial history and genetics [2], ethnicity [3], aging [4] and obesity [5]. (lines 63-64)

In families with a history of prostate cancer, the risk of developing the disease doubles in those with a first-degree relationship. In terms of heritability, it appears that predisposition to the disease is due to polygenic mutations [7,8]. (lines 65 – 67)

Numerous studies have highlighted the disparities in prostate cancer incidence across the globe. Incidence of prostate cancer appears to be highest in wealthier countries of the Western world. This is thought to be due to widely available screening and detection processes. Despite these, incidence of prostate cancer remains highest amongst men of African origin [3, 10]. (lines 68 – 72)

Moreover, a study published in 2019 found a strong link between hyperglycaemia and increased risk of fatal prostate cancer [17]. (lines 79-81)

  • “There is a typo error, line 80 a comma instead of a dot.”

Thank you. Corrected to:

imprinting control region (ICR) [24]. (line3 93-94)

Reviewer 2 Report

The manuscript by <kingshott et al described the impact of altered metabolism on the regulation of IGFII/H19 imprinting status in prostate cancer. The manuscript is very well written. The study is well-designed and the methodology is adequate. The results are clearly presented and the conclusions are strongly supported by the results. Moreover the findings are significant for clinical management of prostate cancer patients.

There are only to minor issues to be addressed:

1) the title must be informative of the main finding of the study (i.e Alteration of metabolic conditions disrupt imprinting of....)

2) the abstract is too coincise: please rewrite detaili in particular the amin findings and their possible clinical impact.

Author Response

  • “the title must be informative of the main finding of the study (i.e Alteration of metabolic conditions disrupt imprinting of....)”

Thank you for this suggestion. We have amended the title accordingly, from:

The impact of altered metabolism on the regulation of IGF-II/H19 imprinting status in prostate cancer. (lines 1-2)

To:

Alteration of metabolic conditions impacts regulation of IGF-II/H19 imprinting status in prostate cancer. (lines 2-3)

  • “the abstract is too concise: please rewrite detail in particular, the main findings and their possible clinical impact.”

Thank you for this recommendation. We have amended the abstract to incorporate more detail.

Changed from:

AbstractProstate cancer is the second major cause of male cancer deaths. Obesity, type 2 diabetes, and cancer risk are linked. Insulin-like growth factor II (IGF-II) is involved in numerous cellular events, including proliferation and survival. The IGF-II gene shares its locus with the lncRNA, H19. IGF-II/H19 was the first gene to be identified as being ‘imprinted’ – where the paternal copy is not transcribed – a silencing phenomenon lost in many cancer types. We disrupted imprinting behaviour in vitro by altering metabolic conditions and quantified it using RFLP, qPCR and pyrosequencing; changes to peptide were measured using RIA. Prostate tissue samples were analysed using ddPCR, pyrosequencing and IHC. We compared with in silico data, provided by TGCA on the cBIO Portal.  Disruption of imprinting behaviour, in vitro, occurs at the molecular level with no changes to peptide. In vivo, most specimens retained imprinting status apart from a small subset which showed reduced imprinting. A positive correlation was seen between IGF-II and H19 mRNA expression, which concurred with findings of larger Cancer Genome Atlas (TGCA) cohorts. This positive correlation did not affect IGF-II peptide.  Type 2 diabetes and / or obesity directly affect regulation growth factors involved in carcinogenesis. (lines 38-51)

To include the main findings as below:

Under hyperglycaemic (5mM> glucose) or inflammatory conditions, in vitro, disruption to imprinting occurs, causing a significant (P < .05) increase in IGF-II and a reciprocal decrease in H19 mRNA; the increased mRNA was not translated into peptide. These results were coupled with significant (P < .05) decreases in imprinting percentage. (lines 47-50)

Our findings show that type 2 diabetes and / or obesity can epigenetically affect the regulation of growth factors involved in carcinogenesis. This suggests that modification of lifestyle habits may reduce cancer risk. (lines 53-56)
